# COVID-19-Related Stress and Its Association with Intimate Partner Violence Perpetration and Victimization

**DOI:** 10.3390/bs13070593

**Published:** 2023-07-15

**Authors:** Brooke Keilholtz, Chelsea Spencer, Summer Vail, Megan Palmer

**Affiliations:** Couple and Family Therapy Program, College of Health and Human Sciences, Kansas State University, Manhattan, KS 66506, USA; cspencer@ksu.edu (C.S.); slvail@ksu.edu (S.V.); mpalm1336@ksu.edu (M.P.)

**Keywords:** domestic violence, intimate partner violence, COVID-19, stressors

## Abstract

Intimate partner violence (IPV) is a pervasive issue, and during the COVID-19 pandemic, it has been speculated that the prevalence rates of IPV increased. This paper aims to understand how pandemic-specific distress was related to experiencing and perpetrating IPV. Using self-reported survey data from 371 individuals living in the United States, this study used multiple logistic regressions to examine how reports of distress related to working from home, working outside the home, isolation, stay-at-home orders, mask mandates, physical and mental health, finances, interpersonal relationships, taking care of children, and online learning for children, as well as reports of partner conflict regarding COVID-19, were associated with physical, psychological, and sexual IPV perpetration and victimization. Our results indicated that distress related to family relationships, taking care of children, and COVID-19 as a source of conflict were all associated with an increased risk of IPV victimization, while distress related to mask mandates and friendships was associated with a decreased risk. Distress related to physical health, family relationships, taking care of children, and COVID-19 being a source of conflict were associated with an increased risk of IPV perpetration, while distress related to mental health and friendships was associated with a decreased risk. Implications for researchers and clinicians are discussed.

## 1. Introduction

Intimate partner violence (IPV), the act of causing physical, psychological, and/or sexual harm to one’s current or former partner, is a widespread issue impacting more than 10 million individuals in a given year [1]. IPV is associated with a host of consequences for victims, perpetrators, and those close to them, having the ability to impact nearly all aspects of one’s life, including their physical and mental health, employment and education, housing, relationships with others, and overall quality of life [2,3,4,5]. During the COVID-19 pandemic, there was increased concern about how the pandemic may be heightening rates of IPV [6].

Within the context of the COVID-19 pandemic, there were lifestyle changes that potentially increased the risk of IPV, including social isolation, stay-at-home orders, substance use, mental health issues, and increased stress [7,8,9,10]. When examining data on police reports and calls related to domestic violence from March 2020 compared to March 2019, the rate of reports increased for many counties and cities, with some reports showing increases of 10–27% [11]. It is known that not all instances of IPV are reported to the police, so these numbers may be even higher. It has been hypothesized that the increase in help seeking related to IPV could be related to increased severity of IPV during the pandemic [12]. Due to the increase in IPV cases documented during the COVID-19 pandemic, continued examination of factors specific to the COVID-19 pandemic will be useful to further explore the impact COVID-19 may have had on IPV.

### 1.1. Stressors and COVID-19

The COVID-19 pandemic was stressful time for many individuals and families across the world, causing a major disruption for individuals and families due to school closures, loss of occupation, stay-at-home orders, and isolation [13]. Through these disruptions, increased family stress resulted in increased psychosocial stress, particularly among families who were vulnerable prior to the pandemic (e.g., low socio-economic status or differently abled individuals [14]). Existing stressors were exacerbated, and unique stressors were created during this time, including increased stress related to finances, physical health symptoms associated with a COVID-19 diagnosis, mental health symptoms associated with social isolation and quarantine, as well as lifestyle changes associated with working from home, parenting, and family conflicts [15,16,17].

Throughout the COVID-19 pandemic, there was considerable concern surrounding social isolation and long-term effects on mental health symptoms (e.g., anxiety, depression, substance use, loneliness, and suicidal ideation [17,18]. Notably, several early studies on the psychological effects of the COVID-19 pandemic identified increased levels of anxiety, depression, and stress [19,20], as well as post-traumatic stress disorder symptoms [19,20,21,22]. A study of 1210 individuals in China identified that 53% of the individuals reported a moderate to severe impact on their psychological well-being in relation to the COVID-19 pandemic [23]. Fear associated with the COVID-19 pandemic was also associated with depressive symptoms and anxiety within the U.S. [24]. A systematic review found that during the COVID-19 pandemic, the general public reported lower levels of psychological well-being, including higher rates of anxiety and depression [25]. This evidence suggests that COVID-19 was a stressful event for many individuals and their families, causing a significant impact on their day-to-day functioning as well as their overall well-being.

### 1.2. Stressors and IPV

Previous research has found that stress, including relationship distress, mental health distress, financial stress (such as unemployment or low income), and physical health problems, is linked to IPV perpetration and victimization among men and women [26], all of which, as noted earlier, were increased during the COVID-19 pandemic. There are a number of negative symptoms associated with experiencing high levels of stress that could potentially lead to IPV perpetration, such as anger, challenges with regulating negative emotions, and mental health symptoms [27,28,29]. Additionally, IPV victimization may serve as a unique stressor for an individual that could exacerbate the negative symptoms associated with stress. Due to the connection between stress and IPV and the new and intensified stressors associated with the COVID-19 pandemic, this article seeks to examine COVID-19-specific correlates for IPV perpetration and victimization.

### 1.3. Family Stress Theory

The current study utilizes the lens of family stress theory to understand the impact of COVID-19 on families and how this may be connected to the increased risk of IPV during the pandemic [30]. Family stress theory has been used to examine how families respond to significant stressors, such as the loss of employment and income [31], separation and reunification during wartime [30], and, recently, stressors related to the COVID-19 pandemic [31]. In the seminal literature, Hill (1949) conceptualized a dynamic process, initiated by an event or circumstance, that compels the family system to undergo significant change. In the case of COVID-19, sudden-onset instability affected multiple domains, including, but not limited to, employment and finances, mental and physical health, interpersonal relationships, and access to crucial public resources.

Family stress theory posits that the characteristics of consequential change to the family’s equilibrium are shaped by (1) the resources available to each member and (2) how they perceive or define the event (14; Figure 1). Individual resources are a combination of internal and external resources. Internal resources include health, education, and psychological characteristics like self-esteem. External resources are supports that aid in buffering each partner from stress, such as social support, financial security, community programs, and healthcare. The extent to which resources are available to each partner will determine their coping capacity and strategies. When coping is ineffective, relationships reach a state of crisis and dysregulation (30), increasing the risk of IPV.

Finally, the meaning or perception that each partner places on a causal circumstance has an influence on their capacity for managing stress (30). Though appraisal is frequently subjective, experiencing a pandemic triggered a sense of danger for many people who felt that their perception of the event was a matter of survival. In relationships where there was a discrepancy in the perception of COVID-19, this may have been a source of intimate partner conflict.

The current study focuses on individuals’ perceptions of the distress they experienced regarding a range of stressors related to COVID-19 and how these different stressors potentially influenced the risk of IPV perpetration and victimization. The examined stressors ranged from those that have already been linked to IPV but were exacerbated for many during the pandemic, like physical and mental health distress [29], in addition to stressors that were unique to pandemic experience, like stay-at-home orders and mask mandates. Understanding how these stressors relate to IPV perpetration and victimization may provide further insight into the increased prevalence of IPV during this time [32].

### 1.4. Present Study

There has been increased concern and amplified documentation of IPV rates during the COVID-19 pandemic. Despite increased awareness, limited research has examined the risk markers associated with IPV in this unique context. The current study expands on a 2021 study that examined risk markers for IPV perpetration and COVID-19 [33] by examining different variables and risk markers of both perpetration and victimization. Research has indicated that the increased stress during the pandemic may explain the increase in IPV during this time [32], and our study aims to examine what specific stressors may be linked to IPV. We explore COVID-19-specific stressors in addition to existing stressors that may have been exacerbated by the pandemic, including COVID-19-related distress from working inside or outside one’s home, isolation, stay-at-home orders, mask mandates, physical and mental health, finances, interpersonal relationships, taking care of children, and online learning for children, as well as reports of partner conflict regarding COVID-19, in an effort to understand the potential mechanisms by which partner violence expanded in the United States during the pandemic.

## 2. Methods

The current study used cross-sectional data collected in June 2022. Data was collected through Prolific, an online survey platform, where respondents were chosen to participate based on demographics to allow for a nationally representative sample. Eligibility for the study entailed being English-speaking, older than 18 years of age, and living in the United States. Participants were informed about the study and provided written consent prior to completing the survey. Participants were compensated with USD 12 per hour for survey completion. [Blinded]’s institutional review board approved the study.

### 2.1. Participants

The current study utilized a sample of 371 participants who reported being in a relationship and their IPV experiences within that relationship. The participants ranged in age from 18–80 years old. See Table 1 for demographic information.

### 2.2. Measures

Participants responded to questions about IPV perpetration and victimization and COVID-19-related distress, along with several demographic questions that collected information on participants’ age, gender, race, education, income, and relationship status.

### 2.3. IPV

The Conflict Tactics Scale (CTS-2), developed by Stauss and colleagues (1996), was used to measure physical, sexual, and psychological IPV victimization and perpetration. Participants responded to 78 items, scaling from 0 (this has never happened) to 20 (more than 20 times in the past year), regarding how often certain conflict tactics were used by and against one’s partner (e.g., “kicked”, “swore or cursed”) in the past year. Dummy variables, coded 0 (no IPV) and 1 (IPV), were created for the three different forms of violence for both perpetration and victimization. The internal consistency of the measure was acceptable (*α* = 0.89).

### 2.4. COVID-19 Distress

The survey questions were developed to measure COVID-19-related distress from working inside or outside one’s home, isolation, stay-at-home orders, mask mandates, physical and mental health, finances, interpersonal relationships, taking care of children, and online learning for children, as well as reports of partner conflict regarding COVID-19. Participants were asked “On a scale 1–5 (1 = not at all, 5 = very much), how distressing have the following been related to COVID-19/since March 2020—[stressor]”, and then responded to 13 items regarding the level of distress that COVID-19-specific stressors caused them. Participants were also asked “Do you and your partner have similar views of the COVID-19 pandemic”, and responded from 0 (not at all) to 5 (a lot).

### 2.5. Analysis Plan

Data analysis was completed through SPSS v. 27 [34]. The prevalence of demographic characteristics, IPV, and COVID-19-related distress was examined and determined to be within the sample. Additionally, correlations among these factors were provided. We then ran multivariate logistic regressions to examine whether the presence of COVID-19 distress increased the likelihood of IPV perpetration and victimization across multiple types of violence.

## 3. Results

### 3.1. Correlations

Table 2 includes the correlates of all COVID-19 and IPV variables.

### 3.2. IPV Perpetration

Table 3 includes multivariate logistic regressions examining COVID-19 distress as well as physical, psychological, sexual, or any other form of IPV perpetration.

#### 3.2.1. Physical IPV Perpetration

Higher levels of COVID-19 distress related to physical health (OR = 1.63, *p* = 0.01; see Table 3) and taking care of children (OR = 1.82, *p* < 0.01) were significantly associated with an increased likelihood of physical IPV perpetration. Additionally, higher levels of conflict regarding COVID-19 (OR = 1.58, *p* = 0.02) were significantly associated with an increased likelihood of physical IPV perpetration.

#### 3.2.2. Psychological IPV Perpetration

Higher levels of COVID-19 distress related to family relationships (OR = 1.39, *p* = 0.04) and higher levels of conflict regarding COVID-19 (OR = 2.19, *p* = 0.01) were significantly associated with an increased likelihood of psychological IPV perpetration.

#### 3.2.3. Sexual IPV Perpetration

Higher levels of COVID-19 distress related to physical health (OR = 1.51, *p* = 0.01) were significantly associated with an increased likelihood of sexual IPV perpetration, while higher levels of distress related to mental health (OR = 0.66, *p* = 0.02) were significantly associated with decreased likelihood of sexual IPV perpetration. Additionally, higher levels of conflict regarding COVID-19 (OR = 1.51, *p* = 0.02) were significantly associated with an increased likelihood of sexual IPV perpetration.

#### 3.2.4. Any IPV Perpetration

Higher levels of COVID-19 distress related to physical health (OR = 1.37, *p* = 0.02) and family distress (OR = 1.65, *p* < 0.01) were significantly associated with an increased likelihood of any IPV perpetration, while higher levels of distress related to friendships was significantly associated with a decreased likelihood of IPV perpetration occurring. Additionally, higher levels of conflict regarding COVID-19 (OR = 1.79, *p* = 0.02) were significantly associated with an increased likelihood of IPV perpetration.

#### 3.2.5. Insignificant Findings

Distress related to working from home, isolation, stay-at-home orders, mask mandates, finances, romantic relationships, working outside the home, online learning for children, as well as having similar views to one’s partner on COVID-19 were not statistically significant in relation with any type of IPV perpetration.

### 3.3. IPV Victimization

Table 4 includes multivariate logistic regressions examining COVID-19 distress as well as physical, psychological, sexual, and any other form of IPV victimization.

#### 3.3.1. Physical IPV Victimization

Higher levels of COVID-19 distress related to family relationships (OR = 1.79, *p* < 0.01; See Table 4) and taking care of children (OR = 1.85, *p* < 0.01) were significantly associated with an increased likelihood of physical IPV victimization, while higher levels of distress related to mask mandates (OR = 0.69, *p* = 0.03) were significantly associated with a decreased likelihood of physical IPV victimization. Additionally, higher levels of conflict regarding COVID-19 (OR = 1.58, *p* = 0.02) were significantly associated with an increased likelihood of physical IPV victimization.

#### 3.3.2. Psychological IPV Victimization

Higher levels of COVID-19 distress related to family relationships (OR = 1.39, *p* < 0.001) and higher levels of conflict regarding COVID-19 (OR = 2.19, *p* < 0.01) were significantly associated with an increased likelihood of psychological IPV victimization. Higher levels of COVID-19 distress related to friendships were significantly associated with a decreased likelihood of psychological IPV victimization (OR = 0.72, *p* = 0.04).

#### 3.3.3. Sexual IPV Victimization

There were no statistically significant findings regarding sexual IPV victimization.

#### 3.3.4. Any IPV Victimization

Higher levels of COVID-19 distress related to family relationships (OR = 1.45, *p* = 0.02) were significantly associated with an increased likelihood of any IPV victimization.

Insignificant Findings

Distress related to working from home, isolation, stay-at-home orders, physical health, mental health, finances, romantic relationships, working outside the home, and online learning for children, as well as having similar views to one’s partner on COVID-19, were not statistically significant in relation to any type of IPV victimization.

## 4. Discussion

As awareness of the increase in IPV during the COVID-19 pandemic is growing, more information is needed to understand what potentially contributed to this increase. One common explanation of this dynamic has been the increased isolation and distress brought on by the pandemic [32]. The current study examined multiple stressors either related or unique to the COVID-19 pandemic and their relation to IPV perpetration and victimization in an effort to examine this potential explanation. It was found that higher levels of COVID-19 distress related to family relationships, taking care of children, and physical health, along with COVID-19 being a source of conflict within the relationship, all significantly increased the risk of one or more forms of IPV being perpetrated, while distress related to mental health and friendships was found to be protective and significantly decreased the risk of IPV perpetration. Additionally, higher levels of COVID-19 distress related to family relationships and taking care of children, as well as reports of COVID-19 being a source of conflict within the relationship, all significantly increased the risk of one or more forms of IPV victimization, while distress related to mask mandates and friendships was found to be protective and significantly decreased the risk of IPV victimization. Our findings support that the increase in and creation of new stressors brought on by the COVID-19 pandemic may assist us in understanding the rise in IPV during this time. Clinical and research implications are addressed below.

In addition to examining the link between these COVID-19 stressors and IPV variables, correlates of all variables were examined. These results indicated that most COVID-19 stressors were associated with one another in a similar way that IPV variables were correlated with one another. Research has confirmed the link between different forms (physical, psychological, and sexual) and directions (perpetration and victimization) of IPV [29,35]. Yet, less is understood about how these COVID-19 stressors relate to one another. Our findings suggest that those experiencing increased distress regarding one aspect/stressor of COVID-19 were likely experiencing distress regarding multiple aspects.

IPV’s association with stress [26], as well as the increase in stressors during the COVID-19 pandemic [15,16,17], are both well-documented. Although nearly all COVID-19 stressors which were examined were correlated with one another, only a few of these stressors were linked with IPV perpetration and victimization, suggesting that it may not be general stress that is linked to violence, but stress related to specific issues. Distress related to family relationships and taking care of children were both found to be linked with an increase in IPV perpetration and victimization. This aligns with previous research that has identified parenting and familial stress as correlates of IPV [36,37]; these factors have been examined as both outcomes and predictors of it. One potential explanation for this is that these stressors are likely navigated within the context of one’s intimate relationship. If partners try to resolve these issues but are feeling distressed by them, they are more susceptible to placing negative intent and blame on one another [38], which increases the risk of IPV [39,40]

Additionally, COVID-19 being a source of conflict in one’s relationship significantly increased the likelihood of IPV perpetration and victimization. IPV’s tie with conflict is well-established [41], yet this finding highlights that social/political issues may cause some relational conflict that leads to IPV. COVID-19 as a source of conflict was negatively correlated with similar partner views, suggesting that this conflict may be related to differences in opinions regarding COVID-19. A difference in political orientation from one’s partner has been linked with lower relational satisfaction [42], and as political issues increase in presence and polarity, we may see them have an increased impact on intimate relationships, potentially increasing the risk of conflict and IPV.

Distress regarding physical health also increased the likelihood of IPV perpetration, but only for physical violence. Poor physical health has been previously identified as a risk factor for physical IPV perpetration [29], potentially explained by the increased distress and decrease in feelings of power it brings, which may ultimately lead individuals experiencing physical health issues to perpetrate violence. Additionally, COVID-19 created an increase in physical health issues for many, potentially leading to increased distress and fear surrounding physical health during this time and making it a more present risk factor.

In addition to factors that increased the risk of IPV, our study identified several protective factors that decreased the likelihood of certain forms of IPV. Distress regarding friendships decreased the likelihood of any IPV perpetration and psychological IPV victimization. One potential explanation for this is that friend-related distress increased the dependency on and desire to manage one’s intimate relationship. Additionally, distress regarding mask mandates decreased the likelihood of psychological IPV victimization, and distress regarding mental health decreased the likelihood of sexual IPV perpetration.

### 4.1. Clinical Implications

It is crucial for clinicians to be aware of the specific COVID-19-related risk markers for IPV. In addition to knowing and recognizing the risks, clinicians should be assessing the degree of distress that each of these stressors applies to their clients and providing psychoeducation to those at risk of IPV. By increasing clients’ awareness of the impact of these stressors, collaborative tracking of the mechanisms by which stress translates to violence may occur and create opportunities for prevention.

In congruence with the family stress theory, individual and family resources should be evaluated and utilized to minimize the risk of experiencing IPV. Teaching coping skills and using cognitive interventions, such as reframing or challenging automatic negative thoughts, may be used as initial methods of increasing internal resources. In addition to connecting clients with available public resources, clinicians often should be involved in advocacy with larger systems to improve families’ access to needed support. Finally, our findings suggest that having social relationships outside of the home, such as friendships, and feeling distress over separation from them is a protective marker against the perpetration and victimization of IPV. This implies that continued efforts to connect with and build social support outside of the home are an external resource that should be prioritized in clinical work.

### 4.2. Research Implications

This study identified that many COVID-19 stressors were associated with one another, highlighting complexities in how we understand and explore stressors related to COVID-19. As noted earlier, despite these stressors being correlates of one another, only some were significantly associated with IPV perpetration or victimization. This finding suggests that there is value in examining specific sources of stress, compared to general distress, when examining factors that may have promoted the increase in IPV during the COVID-19 pandemic. Understanding which specific stressors are associated with an increased risk of IPV allows us to be more accurate in our assessment and prevention measures. Additionally, our results found that many of the stressors that were found be associated with an increased risk of IPV perpetration and victimization were not stressors unique to the COVID-19 pandemic; instead, they were stressors that were enhanced during the pandemic, but may still be impacting individuals and families.

### 4.3. Limitations and Future Research

The first limitation of this study is that cross-sectional data collected in June 2022, asking about experiences in the past year, was utilized. Therefore, the data are retrospective and focused on acts of IPV that occurred between mid-2021 and mid-2022, limiting our ability to make generalizations about the beginning of the COVID-19 pandemic. It would have been useful to have had longitudinal data collected throughout the COVID-19 pandemic in order to examine how COVID-19-related stressors impacted IPV trends over time. Additionally, we do not know whether the COVID-19 stressors were antecedents of IPV, or whether IPV within the relationship worsened the impact of COVID-19-related stressors. In this study, we only examined IPV as a dichotomous variable, which limited our ability to examine the severity and frequency of IPV within the relationship. It may be useful for future research to explore how COVID-19 stressors impact the severity and frequency of IPV. During COVID-19, there was also an increase in resources and programs aimed at lessening or eliminating these stressors, so although some may have experienced more distress in the areas examined in this study, some may also have found relief and decreased distress. Future research may benefit from examining how relief or programs related to COVID-19 influenced the relationship between IPV and COVID-19 stressors.

Another limitation of this study is that the sample consisted of predominantly White individuals living in the US, which limits the generalizability of the results to people of color and individuals not residing in the US. Similarly, the majority of the sample identified as heterosexual, which limits the generalizability of the findings to the LGBT+ community. Research indicates that the risk factors for IPV perpetration and victimization significantly vary in regard to race [43] and sexual orientation, and that there are unique risk factors for these populations as well [44]. Future research would benefit from examining other populations to examine whether COVID-19 stressors had similar or differing impacts on IPV perpetration and victimization among these populations.

Lastly, another limitation was the lack of scale development (e.g., reliability and validity tests) prior to the deployment of this scale to a larger sample pool. The scale used in this study identified 13 areas of potential distress related to COVID-19, but there were many areas of distress at this time, and some were likely not accounted for within this scale. Additionally, different populations likely experienced COVID-19 stressors differently, leading to this scale being more or less relevant for certain populations. Further examination and development of the COVID-19 distress scale used in this study are needed.

## 5. Conclusions

This study aimed to increase our understanding of the mechanisms involved in the rise of IPV during the COVID-19 pandemic. We identified several COVID-19-related stressors, as well as increased conflict regarding COVID-19 within one’s intimate relationship, to be associated with an increased risk of IPV perpetration and victimization, supporting the idea that increased distress during this time may assist in explaining why this rise occurred. Recognition of individual and interpersonal experiences, as well as public programs and policies, may play a critical role in preventing increased violence during the current COVID-19 management period and any future similar emergencies. Future research may expand on these specific stressors, or may examine stressors as they relate to specific identities, socioeconomic positions, or ethnicities. Couple and family therapists and social workers on the front lines of addressing IPV should be aware of these risk and protective factors in an effort to promote IPV prevention and reduce the risk of IPV.

## Figures and Tables

**Figure 1 behavsci-13-00593-f001:**
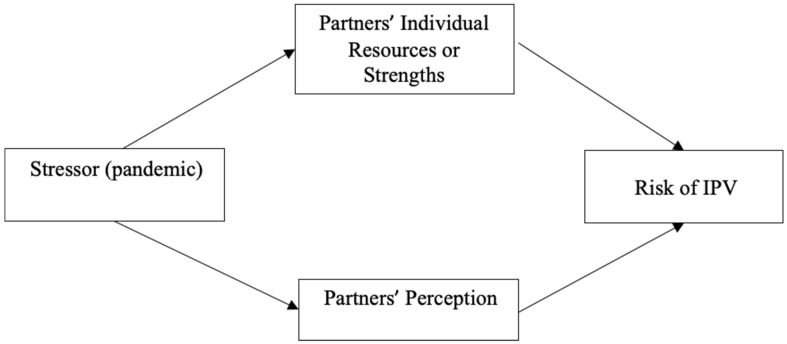
Family stress theory model.

**Table 1 behavsci-13-00593-t001:** Sample demographics and statistics.

	N/M	%/SD
Respondent’s Age	45.05	15.22
Race		
White	288	78
Black	50	14
American Indian	9	2
Asian	23	6
Hispanic	33	9
SW Asian	2	1
Other	4	1
Relationship Status		
Married	185	50
Committed monogamous	109	29
CNM	4	1
Committed poly	1	0
Single	72	19
Sexual Orientation		
Heterosexual	314	85
Gay/Lesbian	14	4
Bisexual	31	8
Pansexual	5	1
Queer	3	1
Other	3	1
Gender Identity		
Male	169	46
Female	196	53
Non-Binary	5	1
Employment		
Employed Full-time	183	49
Employed Part-time	56	15
Unemployed by choice	29	8
Unemployed not by choice	25	7
Retired	57	15
Full Time Student	14	4
Prefer not to say	7	2
Education		
Some High School	5	1
High School Diploma or GED	118	30
Trade School	15	3
Bachelor’s Degree	163	44
Master’s Degree or higher	70	19
Household Income		
Less than USD 25,000	52	14
USD 25,000–50,000	98	26
USD 50,000–100,000	131	35
USD 100,000–200,000	67	18
More than USD 200,000	20	5
Prefer not to say	3	1
COVID-19 Stressors		
Working from home	1.51	0.97
Isolation	2.55	1.30
Stay-at-home orders	2.37	1.31
Mask mandates	2.21	1.33
Physical health	1.97	1.17
Mental health	2.43	1.30
Finances	2.13	1.33
Family relationships	2.12	1.18
Romantic relationships	1.76	1.16
Friendships	2.16	1.16
Working outside the home	1.68	1.13
Taking care of children	1.44	0.95
Online learning for children	1.67	1.18
COVID-19 as source of conflict	1.27	0.70

Note. N = 371.

**Table 2 behavsci-13-00593-t002:** Correlation matrix.

	1	2	3	4	5	6	7	8	9	10	11	12	13	14	15	16	17	18	19	20	21	22
1. Working from home distress																						
2. Isolation distress	**0.42 *****																					
3. Stay-at-home orders distress	**0.37 *****	**0.70 *****																				
4. Mask mandates distress	**0.21 *****	**0.34 *****	**0.61 *****																			
5. Physical health distress	**0.37 *****	**0.50 *****	**0.44 *****	**0.32 *****																		
6. Mental health distress	**0.40 *****	**0.63 *****	**0.51 *****	**0.29 *****	**0.60 *****																	
7. Finances distress	**0.28 *****	**0.39 *****	**0.39 *****	**0.29 *****	**0.51 *****	**0.51 *****																
8. Family relationships distress	**0.33 *****	**0.52 *****	**0.51 *****	**0.33 *****	**0.45 *****	**0.48 *****	**0.50 *****															
9. Romantic relationships distress	**0.33 *****	**0.21 *****	**0.34 *****	**0.25 *****	**0.46 *****	**0.46 *****	**0.46 *****	**0.57 *****														
10. Friendships distress	**0.37 *****	**0.53 *****	**0.46 *****	**0.27 *****	**0.50 *****	**0.50 *****	**0.39 *****	**0.66 *****	**0.59 *****													
11. Working outside of home distress	**0.28 *****	**0.31 *****	**0.31 *****	**0.22 *****	**0.37 *****	**0.34 *****	**0.39 *****	**0.33 *****	**0.30 *****	**0.26 *****												
12. Child-related distress	**0.33 *****	**0.24 *****	**0.23 *****	**0.21 *****	**0.21 *****	**0.24 *****	**0.24 *****	**0.31 *****	**0.36 *****	**0.28 *****	**0.36 *****											
13. Child online learning distress	**0.22 *****	**0.26 *****	**0.27 *****	**0.31 *****	**0.23 *****	**0.27 *****	**0.29 *****	**0.27 *****	**0.33 *****	**0.26 *****	**0.30 *****	**0.53 *****										
14. COVID-19 as source of conflict	**0.24 *****	**0.24 *****	**0.27 *****	**0.22 *****	**0.20 *****	**0.28 *****	**0.17 *****	**0.26 *****	**0.33 *****	**0.24 *****	**0.13 ***	**0.19 *****	0.06									
15. Similar partner views on COVID-19	−0.06	0.03	0.01	−0.06	−0.01	0.04	−0.08	−0.04	**−0.12 ***	0.05	−0.06	**−0.13 ***	−0.03	**−0.17 ****								
16. Physical IPV perpetration	**0.13 ***	0.07	0.09	0.04	**0.14 ****	0.05	0.09	0.14 *	**0.16 ****	**0.14 ****	**0.12 ***	**0.20 *****	0.04	0.20 ***	−0.08							
17. Physical IPV victimization	**0.14 ****	0.10	0.10	0.04	**0.13 ***	0.11 *	0.14 **	0.21 ***	**0.25 *****	0.10	**0.14 ****	**0.24 *****	0.04	**0.25 *****	**−0.12 ***	**0.60 *****						
18. Psychological IPV perpetration	**0.14 ****	0.23 ***	0.18 ***	0.11 *	**0.20 *****	0.22 ***	0.20 ***	0.24 ***	**0.22 *****	**0.16 ****	**0.16 ****	**0.15 ****	0.10	**0.20 *****	0.01	**0.26 *****	**0.30 *****					
19. Psychological IPV victimization	**0.12 ***	0.21 ***	0.17 **	0.09	**0.14 ****	0.17 ***	0.20 ***	0.27 ***	**0.22 *****	**0.15 ****	**0.15 ****	**0.11 ***	**0.12 ***	**0.19 *****	0.03	**0.25 *****	**0.31 *****	**0.84 *****				
20. Sexual IPV perpetration	0.10	0.05	0.11 *	0.14 **	**0.12 ***	0.01	0.12 *	0.17 **	**0.15 ****	0.08	0.06	0.12 *	0.09	**0.14 ****	**−0.11 ***	**0.27 *****	**0.22 *****	**0.20 *****	**0.21 *****			
21. Sexual IPV victimization	**0.12 ***	**0.12 ***	**0.14 ****	**15 ****	**0.16 ****	**0.05 ****	**0.17 ****	**0.17 *****	**0.16 ****	**0.13 ***	**0.17 ****	**0.19 *****	0.10	**0.16 ****	**−0.14 ***	**0.29 *****	**0.33 *****	**0.30 *****	**0.28 *****	**0.61 *****		
22. IPV perpetration	**0.13 ***	**0.20 *****	**0.22 *****	**0.17 ****	**0.24 *****	**0.20 *****	**0.20 *****	**0.31 *****	**0.28 *****	**0.18 *****	**0.18 *****	**0.11 ***	0.08	**0.22 *****	0.01	**0.33 *****	**0.33 *****	**0.74 *****	**0.69 *****	**0.27 *****	**0.43 *****	
23. IPV victimization	0.05	**0.16 ****	**0.17 *****	**0.13 ****	**0.12 ***	**0.16 ****	**0.20 *****	**0.23 *****	**0.19 *****	**0.11 ***	**0.14 ****	**0.11 ***	0.10	**0.15 ****	0.04	**0.33 *****	**0.38 *****	**0.67 *****	**0.75 *****	**0.35 *****	**0.44 *****	**0.82 *****

Note: Boldface indicates statistical significance.* *p* < 0.05; ** *p* < 0.01; *** *p* < 0.001.

**Table 3 behavsci-13-00593-t003:** Logistic regression examining COVID-19 stressors and IPV perpetration.

	Any IPV Perpetration		Physical IPV Perpetration		Psychological IPV Perpetration		Sexual IPV Perpetration	
COVID-19 Stressors	OR	95% CI	OR	95% CI	OR	95% CI	OR	95% CI
Working from home	0.82	[0.61, 1.08]	0.96	[0.65, 1.41]	0.89	[0.67, 1.19]	1.01	[0.72, 1.40]
Isolation	0.97	[0.72, 1.32]	0.84	[0.54, 1.31]	1.13	[0.85, 1.52]	0.83	[0.58, 1.19]
Stay-at-home orders	1.14	[0.84, 1.54]	1.31	[0.84, 2.06]	0.99	[0.74, 1.33]	1.21	[0.84, 1.74]
Mask mandates	1.02	[0.81, 1.28]	0.82	[0.59, 1.15]	0.93	[0.74, 1.16]	1.00	[0.77, 1.32]
Physical health	**1.37**	**[1.05, 1.78]**	**1.63**	**[1.11, 2.40]**	1.20	[0.91, 1.57]	**1.51**	**[1.10, 2.07]**
Mental health	0.96	[0.73, 1.25]	0.73	[0.48, 1.11]	1.09	[0.83, 1.43]	**0.66**	**[0.47, 0.94]**
Finances	0.96	[0.75, 1.21]	0.86	[0.60, 1.24]	1.01	[0.80, 1.29]	1.03	[0.78, 1.38]
Family relationships	**1.65**	**[1.21, 2.25]**	**1.17**	**[0.77, 1.79]**	**1.39**	**[1.02, 1.90]**	1.28	[0.90, 1.82]
Romantic relationships	1.31	[0.97, 1.75]	1.04	[0.72, 1.49]	1.09	[0.81, 1.45]	1.15	[0.85, 1.55]
Friendships	**0.72**	**[0.52, 0.99]**	1.17	[0.77, 1.79]	0.81	[0.60, 1.09]	0.87	[0.60, 1.24]
Working outside of home	1.00	[0.78, 1.29]	0.92	[0.65, 1.31]	1.02	[0.79, 1.32]	0.88	[0.65, 1.18]
Taking care of children	0.96	[0.70, 1.39]	**1.82**	**[1.20, 2.76]**	1.12	[0.77, 1.62]	1.28	[0.89, 1.84]
Online learning for children	1.02	[0.775, 1.34]	0.73	[0.58, 1.11]	1.12	[0.85, 1.49]	0.86	[0.62, 1.19]
COVID-19 as source of conflict	**1.79**	**[1.11, 2.91]**	**1.58**	**[1.09, 2.28]**	**2.19**	**[1.20, 3.98]**	**1.51**	**[1.06, 2.15]**
Similar partner views on COVID-19	1.09	[0.90, 1.32]	0.89	[0.69, 1.15]	1.00	[0.82, 1.21]	0.92	[0.75, 1.14]

Note: OR = odds ratio; CI = confidence interval. Boldface indicates statistical significance.

**Table 4 behavsci-13-00593-t004:** Logistic regression examining COVID-19 stressors and IPV victimization.

	Any IPV Victimization		Physical IPV Victimization		Psychological IPV Victimization		Sexual IPV Victimization	
COVID-19 Stressors	OR	95% CI	OR	95% CI	OR	95% CI	OR	95% CI
Working from home	0.82	[0.82, 1.09]	1.02	[0.71, 1.47]	0.93	[0.70, 1.24]	0.98	[0.72, 1.32]
Isolation	1.05	[0.79, 1.40]	0.74	[0.49, 1.13]	1.19	[0.88, 1.60]	0.86	[0.61, 1.20]
Stay-at-home orders	1.06	[0.80, 1.41]	1.25	[0.82, 1.90]	1.00	[0.74, 1.35]	1.12	[0.80, 1.57]
Mask mandates	1.07	[0.85, 1.33]	**0.69**	**[0.50, 0.97]**	0.89	[0.71, 1.12]	0.99	[0.77, 1.27]
Physical health	0.97	[0.75, 1.25]	1.18	[0.82, 1.69]	1.06	[0.81, 1.39]	1.21	[0.90, 1.61]
Mental health	0.99	[0.77, 1.27]	1.01	[0.69, 1.47]	0.91	[0.69, 1.19]	0.95	[0.70, 1.29]
Finances	1.10	[0.87, 1.39]	1.00	[0.73, 1.38]	1.01	[0.79, 1.29]	1.13	[0.88, 1.47]
Family relationships	**1.45**	**[1.08, 1.94]**	**1.79**	**[1.18, 2.70]**	**1.76**	**[1.27, 2.41]**	1.13	[0.81, 1.57]
Romantic relationships	1.19	[0.91, 1.56]	1.40	[1.00, 1.97]	1.20	[0.89, 1.62]	0.92	[0.69, 1.22]
Friendships	0.75	[0.56, 1.01]	1.17	[0.77, 1.79]	**0.72**	**[0.53, 0.99]**	1.06	[0.76, 1.49]
Working outside of home	1.04	[0.82, 1.33]	0.92	[0.65, 1.31]	1.03	[0.80, 1.33]	1.05	[0.81, 1.36]
Taking care of children	1.05	[0.76, 1.45]	**1.85**	**[1.24, 2.77]**	0.84	[0.59, 1.20]	1.35	[0.97, 1.87]
Online learning for children	1.03	[0.79, 1.35]	0.73	[0.48, 1.11]	1.34	[0.99, 1.80]	0.92	[0.68, 1.23]
COVID-19 as source of conflict	1.37	[0.922, 1.04]	**1.58**	**[1.09, 2.28]**	**2.1**	**[1.21, 3.65]**	1.25	[0.90, 1.74]
Similar partner views on COVID-19	1.11	[0.92, 1.34]	0.89	[0.69, 1.15]	1.03	[0.85, 1.25]	0.84	[0.68, 1.02]

Note: OR = odds ratio; CI = confidence interval. Boldface indicates statistical significance.

## Data Availability

The data that support the findings of this study are available from the corresponding author, B.K., upon reasonable request.

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
