# Peer review of "COVID-19-Related Stress and Its Association with Intimate Partner Violence Perpetration and Victimization"

_behavsci, 2023, doi:10.3390/bs13070593_

Round 1

Reviewer 1 Report

Thank you for giving me the opportunity to read this paper.  Overall I thought the work was clear, coherent and timely; providing some important recommendations to support clinicians who are working with those impacted by IPV during COVID-19.  There were strong links to appropriate theory and academic research and the statistical analysis was systematic and comprehensive.  Further, I particularly liked the use of the Conflict Tactics Scale to measure IPV as this provided data on different IPV dimensions - important in the current context and the diversity of the population (although still skewed towards heterosexual relationships and White participants) begins to shine a light on the complexity of perpetration and victimisation.  As the authors identify in their limitations section, it would be interesting to see further research capturing the perspectives from these groups.  It would however be useful I think to extend this discussion slightly to highlight how this would relate to what is already known about IPV in such samples.  I feel this could be addressed in a further paragraph.

In terms of areas which would benefit from further articulation, I feel it would be useful to know more about the COVID specific scale developed for this study.  Were any assessments for reliability and validity conducted with this scale prior to deployment in this study?  How were the 13 items selected and how large was the original item pool?  Some further detail (it does not have to be much and could involve the authors referring to other work which has been done on scale validation) would be useful. Alternatively if this was not carried out, some reflection on this as a potential limitation in the discussion would be useful. 

The other area where I felt that the argument was underplayed was in the "how we use this going forward".  The authors have addressed this when discussing the implications for clinicians working with these target populations.  However it would be helpful to understand whether these findings could help with "identification" and "prevention" of future IPV or whether the authors feel this is entirely COVID specific.  It would be hoped that the pandemic was a once in a generation occurrence but are there lessons which could be generalised beyond the COVID context to continue to be applicable as we recover from COVID and move beyond this?  If the authors judge that this is a time and context specific study that still gives it merit from the perspective of intervening to promote recovery.  However being more explicit about how generalisable the learning is to wider contexts would strengthen the work.   I think that this might be particularly important as aside from the mask mandates with physical IPV, none of the other ORs were all that COVID specific.  Even the "COVID as a source of conflict" is not that specific - because the authors did not ask about other events as potential sources of conflict.  For that reason, COVID may not have been a source of conflict in itself, merely an excuse for conflict to occur which would have happened anyway.  This is something to consider unpicking a little more.

In terms of the statistics, these were really well presented and the logistic regressions allowed for the complexity of these interrelationships to be brought out.  This was helpful as the number of variables in the bivariate correlations was in danger of muddying the message rather than making this clear .  Perhaps a short narrative around the correlations to contextualise this for the reader would be useful.  However in general the statistics were entirely appropriate and I could see that the authors had followed a purposeful strategy of linking logistic regressions to the predictions with a clear analytic plan. One minor point, results are never "insignificant". They are "statistically nonsignificant".  The term "insignificant" implies "unimportant" and this is not something which is necessarily true of a statistically nonsignificant finding.  Therefore this needs to be altered in the text of the paper. 

Overall I felt that the paper had some clear merits but that there were areas where this could be enhanced further.  I wish the authors success in their future work. 

Author Response

Thank you for giving me the opportunity to read this paper.  Overall I thought the work was clear, coherent and timely; providing some important recommendations to support clinicians who are working with those impacted by IPV during COVID-19.  There were strong links to appropriate theory and academic research and the statistical analysis was systematic and comprehensive.  Further, I particularly liked the use of the Conflict Tactics Scale to measure IPV as this provided data on different IPV dimensions - important in the current context and the diversity of the population (although still skewed towards heterosexual relationships and White participants) begins to shine a light on the complexity of perpetration and victimisation.  As the authors identify in their limitations section, it would be interesting to see further research capturing the perspectives from these groups.  It would however be useful I think to extend this discussion slightly to highlight how this would relate to what is already known about IPV in such samples.  I feel this could be addressed in a further paragraph. –

- We added the following: “Research indicates that risk factors for IPV perpetration and victimization significantly vary in regard to race (Kelly et al., 2022) and sexual orientation, and that there are unique risk factors for these populations as well (Kimmes et al., 2019).”.

In terms of areas which would benefit from further articulation, I feel it would be useful to know more about the COVID specific scale developed for this study.  Were any assessments for reliability and validity conducted with this scale prior to deployment in this study?  How were the 13 items selected and how large was the original item pool?  Some further detail (it does not have to be much and could involve the authors referring to other work which has been done on scale validation) would be useful. Alternatively if this was not carried out, some reflection on this as a potential limitation in the discussion would be useful. –

- We added the following: “Lastly, another limitation was lack of scale development (e.g., reliability and validity tests) prior to deployment of this scale to a larger sample pool. The scale used in the study identified 13 areas of potential distress related to COVID-19 but there were many areas of distress at this time, and some were likely not accounted for within this scale. Addi-tionally, different populations likely experienced COVID-19 stressors differently, leading to this scale to being more or less relevant for certain populations. Further examination and development of the COVID-19 distress scale used in this study is needed.”

The other area where I felt that the argument was underplayed was in the "how we use this going forward".  The authors have addressed this when discussing the implications for clinicians working with these target populations.  However it would be helpful to understand whether these findings could help with "identification" and "prevention" of future IPV or whether the authors feel this is entirely COVID specific.  It would be hoped that the pandemic was a once in a generation occurrence but are there lessons which could be generalised beyond the COVID context to continue to be applicable as we recover from COVID and move beyond this?  If the authors judge that this is a time and context specific study that still gives it merit from the perspective of intervening to promote recovery.  However being more explicit about how generalisable the learning is to wider contexts would strengthen the work.   I think that this might be particularly important as aside from the mask mandates with physical IPV, none of the other ORs were all that COVID specific.  Even the "COVID as a source of conflict" is not that specific - because the authors did not ask about other events as potential sources of conflict.  For that reason, COVID may not have been a source of conflict in itself, merely an excuse for conflict to occur which would have happened anyway.  This is something to consider unpicking a little more. –

- We do try to highlight that many of these stressors are/were existing stressors for many families, they were just exacerbated during the COVID-19 pandemic. But because we ask about how these stressors were specifically experienced during/in relation to the pandemic/after March 2020, we do attempt to be careful of how much we generalize these findings beyond this context. We hope that by connecting these findings to previous research on risk factors of IPV, readers can generalize appropriately.

In terms of the statistics, these were really well presented and the logistic regressions allowed for the complexity of these interrelationships to be brought out.  This was helpful as the number of variables in the bivariate correlations was in danger of muddying the message rather than making this clear .  Perhaps a short narrative around the correlations to contextualise this for the reader would be useful.  However in general the statistics were entirely appropriate and I could see that the authors had followed a purposeful strategy of linking logistic regressions to the predictions with a clear analytic plan. One minor point, results are never "insignificant". They are "statistically nonsignificant".  The term "insignificant" implies "unimportant" and this is not something which is necessarily true of a statistically nonsignificant finding.  Therefore this needs to be altered in the text of the paper. –

- We changed the wording throughout the results to reflect your suggestions.

Overall I felt that the paper had some clear merits but that there were areas where this could be enhanced further.  I wish the authors success in their future work.

- Thank you.

Reviewer 2 Report

1.  Line 10 reports 398 in the sample while lines 149 and 152 report 371.  This discrepancy needs to be resolved/explained.

2.  The authors might consider other sources of stress or relief.  As one example, one of my children decided they didn't want to live in an urban area with strict COVID rules so they gave us two days notice before moving in with us in a rural area, a change that increased our household from two persons to twelve nearly overnight.  We had to double our retirement withdrawals to pay for the food, electricity, laundry, etc. for this increase.  Some of the grandchildren didn't like the idea and yelled at their grandmother to which I took offense (I considered it as emotional abuse) as we were making a lot of sacrifices to accommodate them.  They stayed with us for nine+ months.  Then a judge in Colorado ordered my wife, grandmother, to move to Colorado for eight months without my permission, which I didn't think judges in other states had the power to do.  That meant that sometimes I was left alone to manage the eight grandchildren, ages 2-16, by myself, as the parents were often gone for work or shopping or appointments in other parts of the state.  Not exactly my view of retirement!  On the relief side, it was a huge benefit that local schools provided free lunches for the eight children, saving me a ton or work each weekday, which was a huge relief in many ways.  The main point is that I suspect that we have only begun to tap the ways in which COVID added stress to the lives of families.  Furthermore, my wife broke her leg and was in hospital/rehab for two months during which time she caught COVID from someone in the room next door.  That meant that I would catch it and then be barred from visiting her for nearly three weeks, which didn't improve my morale nor hers.  She didn't get to vote either as I applied too late for an absentee ballot for her.  The point is that there were intersecting stressors.

3.  Line 149, I would prefer to see the range of ages as well as the mean, SD, as a check on data validity.

4.  Line 157.  It needs to be more clear how the split was made between no IPV and "some" IPV - at a mean, median, or some other cut-off point.  What percent fell into each of those two categories?  Of course, at least for both main types of IPV.

5.  Line 165   These measures need better, more precise definition.  What were the questions and the possible responses?  What were the mean scores, SD's, ranges? 

6.  If there have been previous papers published from the same data set, this needs to be clarified more clearly, especially if different sample sizes were used, with changes in sample sizes used clarified.

7.  Table 2 has either some typos or some confusing numbers.  The correlations from items 1 to 13 are pretty straightforward but from there it becomes unusual.  What does 0.0 8 mean?  or 0.1 when all other correlations are reported to two digits?  One correlation of .05 is significant (p < .01) but it seems that some correlations of .10 or so are not significant, which doesn't make sense. 

8.  Line 188 and throughout the results section.  Saying that variable A increased variable B assumes a causal relationship, which cannot be ascertained from correlational, cross-sectional data.  Please rewrite as non-causal types of statements. 

9.  Line 322.  For example, free COVID testing and free school lunches during the summer months, even if children were not in school per se, were very helpful public resources.

10.  Line 439.  Seems to be too much extra spacing on this line.

5. 

Author Response

  1. Line 10 reports 398 in the sample while lines 149 and 152 report 371.  This discrepancy needs to be resolved/explained.

- This was corrected on line 10.

  1. The authors might consider other sources of stress or relief.  As one example, one of my children decided they didn't want to live in an urban area with strict COVID rules so they gave us two days notice before moving in with us in a rural area, a change that increased our household from two persons to twelve nearly overnight.  We had to double our retirement withdrawals to pay for the food, electricity, laundry, etc. for this increase.  Some of the grandchildren didn't like the idea and yelled at their grandmother to which I took offense (I considered it as emotional abuse) as we were making a lot of sacrifices to accommodate them.  They stayed with us for nine+ months.  Then a judge in Colorado ordered my wife, grandmother, to move to Colorado for eight months without my permission, which I didn't think judges in other states had the power to do.  That meant that sometimes I was left alone to manage the eight grandchildren, ages 2-16, by myself, as the parents were often gone for work or shopping or appointments in other parts of the state.  Not exactly my view of retirement!  On the relief side, it was a huge benefit that local schools provided free lunches for the eight children, saving me a ton or work each weekday, which was a huge relief in many ways.  The main point is that I suspect that we have only begun to tap the ways in which COVID added stress to the lives of families.  Furthermore, my wife broke her leg and was in hospital/rehab for two months during which time she caught COVID from someone in the room next door.  That meant that I would catch it and then be barred from visiting her for nearly three weeks, which didn't improve my morale nor hers.  She didn't get to vote either as I applied too late for an absentee ballot for her.  The point is that there were intersecting stressors.

- We added the following “During COVID-19 there was also an increase in resources and programs aimed at less-ening or eliminating these stressors, so although some may have experienced more distress in the areas examined in the study, some may have found relief and decreased distress. Future research may benefit from examining how relief or programs related to COVID-19 influenced the relationship between IPV and COVID-19 stressors.”

  1. Line 149, I would prefer to see the range of ages as well as the mean, SD, as a check on data validity.

- We added the follow: “Participant ranged in age from 18-80 years old.”

  1. Line 157.  It needs to be more clear how the split was made between no IPV and "some" IPV - at a mean, median, or some other cut-off point.  What percent fell into each of those two categories?  Of course, at least for both main types of IPV.

- IPV was dummy coded so either participants were identified as using IPV or not using IPV. There was no use of “some”.

  1. Line 165   These measures need better, more precise definition.  What were the questions and the possible responses?  What were the mean scores, SD's, ranges?  -

- We added the following: “Participants were asked “On a scale 1-5 (1=not at all, 5 = very much” how distressing have the following been related to COVID-19/since March 2020 – [stressor]” and responded to 13 items the level of distress COVID-19 specific stressors caused participants. Participants were also asked “Do you and your partner have similar views of the COVID-19 pan-demic”, and responded from 0 (not at all) to 5 (a lot).” As well as including Means and SDs in table 1.

  1. If there have been previous papers published from the same data set, this needs to be clarified more clearly, especially if different sample sizes were used, with changes in sample sizes used clarified.

- Other papers will be published from this data set but nothing yet from the COVID-19 information. This is the first paper examining risk markers for COVID-19 from this dataset.

  1. Table 2 has either some typos or some confusing numbers.  The correlations from items 1 to 13 are pretty straightforward but from there it becomes unusual.  What does 0.0 8 mean?  or 0.1 when all other correlations are reported to two digits?  One correlation of .05 is significant (p < .01) but it seems that some correlations of .10 or so are not significant, which doesn't make sense. –

- Numbers were corrected to have more consistent format.

  1. Line 188 and throughout the results section.  Saying that variable A increased variable B assumes a causal relationship, which cannot be ascertained from correlational, cross-sectional data.  Please rewrite as non-causal types of statements. 

- Wording was changed throughout the results to reflect your feedback.

  1. Line 322.  For example, free COVID testing and free school lunches during the summer months, even if children were not in school per se, were very helpful public resources.
  2. Line 439.  Seems to be too much extra spacing on this line. –

- Spacing was corrected.